# Beneficial and Detrimental Effects of Cytokines during Influenza and COVID-19

**DOI:** 10.3390/v16020308

**Published:** 2024-02-18

**Authors:** De Chang, Charles Dela Cruz, Lokesh Sharma

**Affiliations:** 1College of Pulmonary and Critical Care Medicine of Eighth Medical Center, Chinese PLA General Hospital, Beijing 100028, China; changde@301hospital.com.cn; 2Department of Pulmonary and Critical Care Medicine of Seventh Medical Center, Chinese PLA General Hospital, Beijing 100028, China; 3Division of Pulmonary, Allergy, Critical Care, and Sleep Medicine, Department of Medicine, University of Pittsburgh, Pittsburgh, PA 15213, USA; delacruz@pitt.edu; 4Veterans Affairs Pittsburgh Healthcare System, Pittsburgh, PA 15240, USA

**Keywords:** Influenza, COVID-19, SARS-CoV-2, Cytokines

## Abstract

Cytokines are signaling molecules that play a role in myriad processes, including those occurring during diseases and homeostasis. Their homeostatic function begins during embryogenesis and persists throughout life, including appropriate signaling for the cell and organism death. During viral infections, antiviral cytokines such as interferons and inflammatory cytokines are upregulated. Despite the well-known benefits of these cytokines, their levels often correlate with disease severity, linking them to unfavorable outcomes. In this review, we discuss both the beneficial and pathological functions of cytokines and the potential challenges in separating these two roles. Further, we discuss challenges in targeting these cytokines during disease and propose a new method for quantifying the cytokine effect to limit the pathological consequences while preserving their beneficial effects.

## 1. Introduction

Cytokines are chemical messengers that signal the growth, maturation, differentiation, and response of cells to various cues during development and physiological functions [1,2]. The essential role of cytokines begins in embryogenesis and persists through the lifetime of an organism. Beyond their homeostatic functions, cytokines also play an instrumental and indispensable role in responding to both infectious and noninfectious threats, as well as aiding in the restoration of homeostasis post-injury [3]. During an active infection, both structural and resident immune cells detect invading pathogens through various pathogen-specific molecules present in their structure, known as pathogen-associated molecular patterns (PAMPs), to upregulate the expression of various cytokines. These cytokines then serve as signals to prime the local structural cells and recruit immune cells for both innate and adaptive immune responses. During a successful and timely pathogen clearance, these initially recruited cells with the help of the structural cells are sufficient to eradicate the pathogen. During non-infectious injuries, the host senses tissue damage through cellular components that are associated with tissue damage known as damage-associated molecular patterns (DAMPs). Sensing this damage leads to the release of cytokines that promote the recruitment of immune cells, leading to tissue repair. Even during infectious diseases, pathogen clearance, especially that mediated by immune cells, often comes at a cost of collateral tissue injury. In addition to the pathogen-targeted host response, the pathogen-mediated tissue damage also serves as a source of DAMPs. Sensing this damage leads to a reparative cytokine response that helps to restore tissue homeostasis by repairing the injured tissue. However, in the absence of effective pathogen clearance, the recruited cells further amplify the cytokine response by recruiting even more immune cells, making the tissue environment highly inflammatory, reaching pathogenic levels. This pathogenic upregulation of cytokine response is often referred to as a “cytokine storm”. Currently, the precise definition of cytokine storm is lacking and so is our knowledge regarding the specific cytokines and associated mechanisms that mediate the pathological events during a cytokine storm. Furthermore, it remains unclear to what extent a reduction in the cytokine levels can safely inhibit a cytokine storm without affecting the host’s ability to clear the pathogen and limiting other beneficial effects of cytokines in the pathogen clearance and subsequent restoration of homeostasis. In this review, we will focus on the role of cytokine response during acute respiratory infections, with a specific emphasis on influenza and COVID-19, to provide a critical overview of how to distinguish their beneficial effects from detrimental effects and explore potential opportunities and challenges in targeting the cytokine storm.

## 2. The Burden of Respiratory Viral Infections

Respiratory viral infections are among the most common causes of infection-mediated morbidity and mortality. From early life, humans experience multiple respiratory infections every year, the majority of which are non-life threatening and do not require medical attention. This is attributed to a well-developed immune system in the lung that is refined by continuous exposure to pathogens and other environmental irritants, such as pollutants, particles, and pathogen remnants.

Despite this well-developed immune system, respiratory infections remain one of the leading causes of death worldwide, resulting in more than half a million deaths annually [4]. The morbidity and mortality by respiratory viruses can reach extraordinary levels, as seen in the current COVID-19 pandemic and during the 1918 influenza pandemic [5,6]. Even in the absence of a pandemic, influenza and related lung diseases were among the top 10 causes of mortality in the United States, indicating their persistent threat. Although the incidence of other viral infections including influenza decreased dramatically due to the onset of COVID-19 and associated preventive measures [7,8], they reemerged as life returned to normal, with the current flu season seeing extensive infections as the total number of infections reached over 4 million in the USA alone this year [9]. The influenza virus features a segmented negative-strand single RNA genome, allowing for reassortment and increased genetic diversity. In contrast, SARS-CoV has a positive-strand single RNA genome, enabling direct translation of viral proteins upon entry into host cells. This distinction in genome structure influences the replication mechanisms and genetic characteristics of these respiratory viruses. Additionally, COVID-19 has emerged as one of the leading causes of death around the world, with the excess mortality due to COVID-19 exceeding three times the reported numbers [5]. The high mortality continues to persist despite widespread vaccinations and improvements in the clinical management of the disease, including the availability of antiviral agents [10]. Additionally, COVID-19 has contributed to mortality from other causes due to lack of appropriate medical care [11]. These cases warrant a deeper understanding of pathophysiological mechanisms that mediate severe or lethal disease in susceptible individuals. In this review, we will focus on cytokine response during respiratory viral infections to understand both challenges and opportunities in targeting them.

## 3. Cytokines during Respiratory Infections

During infections, including respiratory infections, the levels of cytokines are significantly upregulated and can reach as high as nanogram/milliliter levels during severe disease [12,13,14]. However, aging often dampens the initial upregulation of cytokines in response to pathogens, which allows the pathogen to grow uncontrollably [15,16,17], followed by a persistent elevation in cytokine levels that is the hallmark of severe disease. The specific cytokines that are impaired include both interferons, such as IFNβ, and other cytokines, such as IL-2. This has been replicated in the animal models of aging-induced disease severity [18]. At this stage, clinicians often aim to decrease this cytokine response using therapeutic interventions such as corticosteroids or trying to blunt the immune response using specific neutralizing antibodies. These approaches are often effective in diseases mediated by specific cytokine upregulation; however, during infectious diseases, inhibition of cytokines often results in mixed outcomes. Despite their pathological elevation and known pathogenic roles of cytokines, it remains unclear why approaches to inhibit them often fail. This paradox may be explained by the potential beneficial effects of the cytokines, even at pathological levels, in both pathogen clearance and tissue repair (Figure 1).

## 4. Role of Cytokines during Respiratory Infections

In the immune system, cytokines serve as key mediators of immune cell generation, expansion, activation, migration, and functional activation of immune and structural cells. Even at baseline, humans and other mammals maintain measurable cytokine levels that can range from femto- to picomolar levels [19]. The precise baseline functions of these cytokines remain unknown; however, they are believed to be maintaining the homeostatic immune tone. Significantly, for most people, the baseline levels and their relationship with health remain unknown. These cytokines tend to be higher at baseline with aging, especially cytokines such as CRP and IL-6, which correlates with the degree of frailty, indicating a negative impact of cytokines on health even at very low levels [20]. However, some of these cytokines may be dispensable at baseline given the viability and normal life span of various cytokine-knockout mice. It also indicates that the baseline release of cytokines is either to keep the specific transcription machinery active or a response to low pathogenic and environmental exposures throughout life. However, their important roles, both beneficial and pathogenic, become clearly manifested during active infections (Figure 2).

### 4.1. Pathogenic Roles of Cytokines

Our clinical interest in cytokine biology arises from the association of elevated cytokines with severe diseases. This association is conserved in a wide range of pathological conditions including autoimmunity, infections, and non-infectious inflammatory pathologies. However, our understanding of how these cytokines and the specific mechanisms by which they exert deleterious effects on human health is still evolving. Elevated levels of cytokines have both acute and prolonged deleterious effects on human health, as described below.

#### 4.1.1. Acute Pathogenic Roles of Cytokines

Cytokine receptors are expressed in most of the cells in the human body, making them responsive to both the physiological and pathological effects of the cytokines. During severe respiratory viral infections, which are manifested by acute lung injury, cytokines act on the lung endothelium to increase the capillary permeability. The purpose of this increased permeability is to ensure that inflammatory cell recruitment takes place in a timely manner to eradicate the invading pathogens. However, excessive inflammation leads to the accumulation of protein-rich fluid in the alveolar space impairing the gas exchange. An extensive increase in the capillary permeability can lead to extensive extravasation of fluid in peripheral organs leading to sepsis and septic shock, which are manifested by decreased blood pressure and compensatory tachycardia. In addition to the fluid leak, pulmonary upregulation of cytokines can lead to extensive immune infiltration in the lung, which can not only impair the gas exchange but also further amplify the cytokine response by recruiting activated immune cells in the lung parenchyma and alveolar space.

Other than recruiting immune cells, cytokines can directly exert cytotoxic effects. A recent elegant study demonstrated that cytokines TNFα and IFNγ synergistically mediated cell death in immune cells such as macrophages [21]. Similarly, the association of IL-1β release and cell death through pyroptotic mechanisms has been well established. However, cytokines can also have other adverse effects, such as activation of coagulopathy [22], which is associated with severe disease and lethality in viral infections, including during influenza and COVID-19 [23,24,25]. SARS-CoV-2 variants possess unique characteristics that distinguish them from the original or parental virus, often involving mutations in key viral proteins. When assessing the Omicron variant, it is important to note that its potential to induce cytokines, as observed in earlier variants such as Alpha and Delta, requires careful examination. The variations in the genetic makeup of the Omicron variant may influence its ability to stimulate cytokine responses, impacting the severity of the immune response and clinical outcomes. Further research and analysis are necessary to elucidate the specific immunological effects of the Omicron or new variants and their potential role as an inducer of cytokines. Further, some antiviral cytokines such as type I interferons impair the host response against secondary bacterial infections [26], which exacerbates the tissue injury, increasing the overall morbidity and mortality. Of significance, a large proportion of severe patients in both influenza and COVID-19 groups are those with secondary bacterial infections [27,28,29]. The specific details of the host response and the underlying mechanisms involved in the interference with SARS-CoV-2 replication during coinfection should be carefully examined. Additionally, understanding the dynamics of viral-viral coinfections is crucial for elucidating potential interactions between different respiratory viruses and informing public health strategies for managing dual infections [30]. Other than these obvious effects, cytokines can exert multiple other acute effects, including high fever, chills, and hallucinations, which may be markers of cytokines increasing the permeability of the blood–brain barrier or swelling of brain meninges.

#### 4.1.2. Persistent Effects of Cytokines Storm

After the resolution of an acute infection, the cytokine levels quickly fall and reach close to the baseline levels; however, in some conditions they may remain elevated for prolonged periods. During post-acute sequelae of COVID-19, persistent activation of immune response is observed. This immune activation was associated with elevated levels of cytokines such as IL-4, IL-10, and IL-17 [31]. Of significance, these cytokines are not typical proinflammatory cytokines, but fall in the category of Th2 or Th17 cytokine, a phenomenon most likely associated with a prolonged reparative phenotype. Similarly, a prolonged elevation of TGFβ, a major reparative cytokine, can lead to tissue fibrosis, a major concern following inflammatory lung diseases (Figure 2). Of significance, these fibrotic consequences of respiratory viral infections such as influenza and COVID-19 are shown in epidemiological studies and can be replicated in mouse models [32,33,34].

Currently, it remains unknown if other chronic signs of viral infection following recovery are contributed to by elevated cytokine levels. However, the extensive presence of this “long COVID-19” in women [35] may indicate a role of robust immune response, including a cytokine response present in females. This same robust immune response in females may provide them benefits while dealing with an acute infection, as evident by decreased severe disease and mortality in females compared to males across various respiratory infections, including upper and lower respiratory infections [36]. Similar elevation in cytokine response may explain increased vaccine effectiveness and excessive local and systemic side effects in females [37]. During influenza vaccination, females reported increased side effects to the vaccine, including both local (pain, swelling, and redness) and systemic (fatigue, joint pain and other body pain). It is not surprising that females reported elevated antibody titers regardless of influenza vaccine dosing [38]. The biological basis of these elevated cytokine responses is not entirely clear, but mechanistic studies have demonstrated that female antigen-presenting cells such as dendritic cells have an increased ability to respond by producing excessive type I interferon and TNFα response [39]. In contrast to these proinflammatory cytokines, female immune cells produce significantly fewer anti-inflammatory cytokines such as IL-10 in response to influenza infection or a TLR8 ligand [40]. This evidence directly implicates inflammatory/anti-inflammatory cytokines in host immunity, including the humoral response. Our study of COVID-19 subjects demonstrated that age is associated with increased anti-inflammatory cytokine IL-10 levels and the impaired onset of humoral response, including IgG and IgM [16]. However, these young/middle-aged subjects who can clear the acute infection effectively may have chronic consequences, such as post-acute chronic sequelae of COVID. It is important to note that unlike the acute susceptibility to COVID-19, most patients presenting with chronic symptoms are not those who are older (>65 years), but rather middle-aged patients, a population likely to have a robust cytokine response.

Similarly, children are largely protected from acute COVID-19 due to a robust early immune response, including a robust inflammatory responses mediated by type I interferon response and other inflammatory cytokines [41]. However, despite the effective viral clearance, which is evident by no or very limited acute disease, children have increased sequelae of this robust inflammatory response, known as the multisystem inflammatory syndrome in children (MIS-C) [42].

Other prolonged effects of viral infections include cardiac events, including lethal cardiac arrests that contribute to mortality [43]. Other effects may be lung function decline, which may be more evident in patients with preexisting lung diseases such as COPD [44].

## 5. Beneficial Effects of Cytokines

As the detrimental effects of cytokines are well described in both human studies and experimental disease models, including in cells and animals, targeting them should be a rational approach to limit the pathologies. However, efforts to inhibit these cytokines have obtained mixed results, where they often fail to improve the clinical outcome. This is not particularly surprising, given the many essential roles played by cytokines during pulmonary infections.

### 5.1. Antiviral Activity of Cytokines

Interferons are the classical antiviral cytokines and derived their name from their ability to interfere with viral replication [45]. After their initial discovery, a multitude of interferons have been discovered; however, mammals have three prominent classes of interferons named type I, II, and type III, based on their order of discovery. Both type I and type III interferons have a prominent antiviral function [46,47,48] (Figure 3). These interferons stimulate a wide range of genes collectively known as interferon-stimulated genes, which function to inhibit various parts of the viral life cycle. Of significance, their antiviral activities are conserved and are effective against a broad range of viral pathogens, including respiratory pathogens such as influenza and SARS-CoV-2. In addition to classical antiviral cytokines, many other cytokines that are not usually considered antiviral cytokines affect the host’s ability to clear the infecting viruses. Interleukin 6, one of the prominent inflammatory cytokines, is essential for viral control in influenza infection [49]. Others have shown that IL-6 promotes neutrophil survival, which plays a role in viral elimination [50]. Similar antiviral responses have been observed for cytokines like IL-12 [51] and IL-17 [52], both of which contribute to the antiviral response. This evidence shows that the targeting cytokines can impair the host’s ability to clear the viral infection, which can be counterproductive for overall health.

Despite the known antiviral roles of cytokines, including interferons, their biological usefulness, especially during a severe or lethal disease, remains unclear. Although a few smaller clinical studies have shown the benefits of administrating type I interferons in COVID-19 [53], the timing of the therapeutic administrations may be of critical importance. In COVID-19, type I interferon response is associated with both protective [17,54] (when elevated earlier) and detrimental [55] (at the later stage) effects. Multiple lines of evidence indicate that early upregulation of interferon response is associated with better outcomes [17,41]. However, during the later stage of the disease, severe disease is characterized by elevated cytokine levels, including those of antiviral cytokines such as type I interferons [13,55]. This may be due to their effects on the biological processes that are independent of antiviral activities and the clearance of the virus at the later stage of the disease, making their presence unnecessary and harmful. Interferons are known to potentiate host inflammatory response.

### 5.2. Anti-Inflammatory Activity of Cytokines

Elevated levels of cytokines are associated with severe disease in infections; however, the precise function of cytokines during this severe disease phase is not known. It is difficult to rule out that even at pathologically high levels; cytokines have beneficial effects for the host. Interleukin-6, a potent inflammatory cytokine, has been shown to limit the pathogenic inflammatory response in the lung during influenza infection in mice [49]. This indicates that a seemingly proinflammatory cytokine may have anti-inflammatory activities, although one should be careful interpreting these results and should consider the direct effect of IL-6 on viral clearance. Similarly, interleukin-1 deficiency ameliorates the initial inflammatory response to influenza infection; however, this deficiency leads to enhanced death by impairing humoral response [56]. TNFα, another prominent inflammatory cytokine, is not required for viral clearance; however, in its absence, the host fails to regulate the pathological inflammatory response in H1N1 infection [57]. Further, during lethal lung infections, TNFα mediates the pathogenic response and its inhibition provides benefit to the host across multiple strains, including H1N1 [58] or highly pathogenic A/Hong Kong/483/97 strain [59]. These conflicting data show that while absolute elimination or inhibition of certain cytokines may be detrimental to the host, such as seen in knockout mice, their inhibition can potentially provide important therapeutic opportunities. Interleukin-10, a well-known anti-inflammatory cytokine, plays a detrimental role during influenza infection. Its absence not only improves the viral clearance but also limits the inflammatory interferon γ response in mice with influenza infections, indicating a proviral and proinflammatory role [60]. An absence of IL-10 at the time of primary infection leads to enhanced local virus-specific antibody production [60]. This evidence demonstrates the complexity of cytokine response during influenza infections, where the well-known proinflammatory cytokines play a robust anti-inflammatory role along with promoting viral clearance. However, these studies are performed in mouse models where individual cytokines can be targeted by genetic or pharmacological approaches. During infections, many of these cytokines are upregulated together and they further induce expression of each other. It is often difficult to ascertain how much of the pathology is directly contributed to by cytokines themselves and how much by other factors, including direct pathogen-mediated damage.

### 5.3. Role of Cytokine in Tissue Repair

Cytokine release and immune cell recruitment are consistent responses to tissue injury, regardless of the origin of the injury [61,62]. Cytokines such as TGFβ are well described in the literature for their reparative and proliferative actions associated with pathologies such as idiopathic pulmonary fibrosis. TGFβ has been shown to act as a proviral factor in lung epithelium, where it promotes viral replication, and epithelium-derived TGF beta acts to suppress early IFN beta responses, leading to increased viral burden and pathology [63]; however, during appropriate viral replication control, TGFβ acts as an anti-inflammatory cytokine that limits influenza-induced lung pathology [64]. Pulmonary fibrosis has been reported in those recovering from COVID-19, and the incidence seems to be related to the initial disease severity [65]. Another well-known reparative cytokine, fibroblast growth factor 2, is required for pulmonary epithelial injury in a bleomycin model [66]. During influenza infection, fibroblast growth factor 2 limits tissue injury, and its neutralization exacerbates tissue pathology and viral load [67]. Inflammation-resolving processes such as efferocytosis, a mechanism by which macrophages remove the dead immune cells and cellular debris, are associated with the release of anti-inflammatory and tissue-reparative cytokines such as IL-4 and IL-13 [68]. More recent evidence has shown that both type I and type III interferons can inhibit tissue repair in influenza-induced lung injury [69]. Inhibition of infection by interferons while limiting tissue injury repair warrants careful consideration of timing during any potential intervention during viral infections.

## 6. Complexity in Interpreting Cytokine Response

As mentioned above, cytokines play conflicting roles during viral infections, and their effects on the early phase (viral replication) are often different from those exerted at the later phase. Further, it is often observed that despite their upregulation to pathological levels, they often fail to achieve their designated functions, such as viral clearance. The precise mechanisms by which these cytokines exert potentially beneficial and detrimental effects are not known. However, a few factors can drive these effects.

### 6.1. Cytokine Response to the Pathogen vs. Pathogen-Mediated Tissue Damage

The host response in terms of cytokine release can be mediated either in response to the pathogen itself or pathogen-mediated tissue damage. The early upregulation of the cytokines comes from the first mechanism and is often effective in clearing the invading pathogen. This theory is supported by recent evidence that young and healthy patients who can upregulate certain cytokines such as IL-2, even prior to symptomatic disease, are more likely to remain asymptomatic [16]. Further, in older subjects, patients who fail to clear influenza have a more severe inflammatory response and worse clinical outcomes [70]. Similar responses have been shown regarding interferon signaling when persistent interferon signaling is associated with severe disease [55]. However, the late stage of the cytokine response during viral infections is likely related to that caused by tissue damage and may be more detrimental, as they perpetuate the inflammatory and damage signaling in the absence of an external pathogen. Some pathogens might induce early injury to subvert this PAMP-mediated immune response to a damage-related immune response, a phenomenon called immune diversion [71]. Similarly, inflammasome activation, which is a known pathological mechanism in influenza and COVID-19 [32,72], has been shown to be beneficial for generating adaptive immune response [73]. An ideal approach would be to target the host inflammatory signaling induced by tissue damage while protecting the cytokine response to the pathogen itself. Although, this approach appears rational, our inability to precisely distinguish between the cytokine response that is caused by pathogen or tissue damage limits our ability to selectively modulate them.

### 6.2. Functional Saturation of Cytokines

Cytokines are essential for both pathogen clearance and restoring homeostasis after viral infections. However, during the pathogenic elevation of these cytokines, it remains to be known how much signaling of that particular cytokine is required to maintain its physiological/beneficial effects and the excess of these cytokines can be inhibited. This is an important knowledge gap that needs to be filled before cytokine inhibition therapies can be safely applied across infectious diseases. Mere elevation from the baseline should not be used as a justification for their inhibition and the level of inhibition should be determined by measuring the cytokine response in those who are successfully able to fight off infections without exhibiting severe disease. Excess levels of cytokines could then be identified and inhibited to limit the pathological effects of these cytokines while protecting their beneficial functions. Further, it will be important to identify whether the beneficial effects of cytokines have an upper limit and whether they reach saturation at certain levels.

## 7. Cytokine Inhibition: Opportunities and Limitations

Given the pathogenic and persistent elevation in cytokines during acute disease, it is imperative to target those cytokines to limit pathogenic inflammation. However, despite significant elevation in cytokine levels, approaches targeting specific or generic cytokine responses during respiratory viral infections have not always been successful. Influenza causes significant morbidity and mortality during flu seasons and pandemics, especially in older populations. Elevated levels of cytokines are one of the classical hallmarks of severe disease in both influenza and COVID-19, with patterns that are common as well as unique to each infection [74]. Inhibition of cytokine response using corticosteroids has been shown to be detrimental during influenza infection, increasing the duration of the hospital stay, susceptibility to secondary bacterial infections, and overall mortality [75,76]. However, in hospitalized subjects, a lower dose of corticosteroids has been shown to improve the clinical outcomes, including a reduction in mortality in patients with PaO_2_/FiO_2_ < 300 mmHg, a response that was absent in higher-dose or milder patients [77]. These data suggest that complete inhibition of cytokine response may not be useful, while partial inhibition, especially in severe patients, may be beneficial by limiting excessive cytokine response and promoting beneficial cytokine response. Similarly, during COVID-19, cytokine inhibition therapies produced mixed results. Inhibition of IL-6 has shown some clinical benefits in reducing disease severity and mortality; however, randomized clinical trials fail to show any obvious benefit on survival [78,79]. Tocilizumab may lead to side effects such as elevated liver enzymes, gastrointestinal disturbances, headaches, hypertension, changes in blood cell counts, infusion reactions, and an increased risk of serious infections [80]. Similarly, glucocorticoids have been shown to improve the clinical outcomes of COVID-19 [81,82]; however, this improvement was limited to those who were on supplemental oxygen. There was no apparent benefit in those not on ventilators and there was a trend toward worsening outcomes in this population [82]. This is in line with other inflammatory conditions such as sepsis, where low-dose corticosteroids may improve the outcome [83]. Together, this evidence points out that partial inhibition of cytokines, especially in those with severe disease, may provide beneficial effects. In contrast, blocking cytokine response in those with either a mild disease or a complete blockage of cytokine response in those with severe disease may be detrimental, given the beneficial role of cytokines in lung injury repair.

### The Persistent Presence of Cytokines

Although the focus of the cytokine storm remains during the acute disease, a prolonged presence of these cytokines can result in detrimental effects. Tissue remodeling by elevated reparative cytokines such as TGFβ has been well described in the literature. Inflammatory cytokines remain elevated even months after the resolution of acute disease [84]. Recent evidence has indicated that the prolonged presence of certain cytokines such as TNFα, IL-16 and IL-1β is associated with the persistent symptoms of disease in COVID-19 known as post-acute sequelae or long COVID-19 [85]. Through analyses of existing datasets, this study identified macrophages as a major contributor to the cytokine response. Inhibiting the persistent effect of cytokines is another challenge, as the cells that secrete cytokines remain active and their elimination remains a challenge.

## 8. Objective Assessment of Deleterious Effects of Cytokine Storm

Despite the elevation in cytokines during acute disease and their persistence during chronic pathologies, there are limited tools available to quantify the cytokine response that can potentially dictate their excess levels providing quantitative opportunities to inhibit them. Here, we propose a framework for quantitative measures to assess the potentially deleterious effects of cytokines that is adapted from Haber’s rule regarding exposure to toxic chemical gases.

### Haber’s Modified Rule

Fritz Haber was an (in)famous German chemist who contributed to the development of chemical warfare agents such as chlorine. While studying the toxic effects of gases, Haber proposed that the total amount of exposure, which is the multiplication of gas concentration and time of exposure, dictates the overall adverse outcomes [86]. Exposure to a high concentration of toxic gas for a short time has the same outcome as prolonged exposure to a low concentration. However, this rule can be violated at extremely high concentrations, where toxicity and death appear early or at extremely low concentrations, such as environmental pollution where the effects are minimal.

A similar principle can be extended to the cytokine response during various inflammatory diseases, including pulmonary infections. The overall effect of cytokine surge is dictated by both the duration of the surge and the extent of cytokine levels (concentration x time). An extreme elevation can be lethal even if it is elevated for a short duration, such as during septic shock. On the other hand, a prolonged elevation can lead to persistent lung injury resulting in ARDS and subsequent death. As for Haber’s rule, which can be violated at extremes of the concentration, persistent low-grade elevation of cytokines can lead to other pathological conditions, such as pulmonary fibrosis. However, the complexity arises from the fact that unlike toxic gases, cytokines play important homeostatic and tissue-reparative roles during viral infections. Despite these limitations, this rule provides an important framework to limit the deleterious effects of cytokines and provide quantitative measures. However, to make this rule clinically relevant, we need to know the safe concentrations and durations of cytokine upregulation, which remain not well explored.

## 9. Conclusions

The concept of cytokine storms mediating the disease severity and death is not new, and several attempts to mitigate cytokine storms achieved limited evidence of success. Despite the continuous effort, the appropriate cytokine targets and appropriate levels of inhibition have been challenging. This challenge lies in our limited ability to distinguish the beneficial effects of cytokines from the detrimental effects. Cytokines, in addition to playing a vital role in viral clearance, contribute to tissue repair, and both processes can be hampered by therapeutic approaches that inhibit cytokines. Further, inhibition of cytokines can render the host susceptible to secondary bacterial infections, which are major contributors to overall morbidity and mortality. Of significance, a viral pathogen often promotes the host’s susceptibility to secondary bacterial infections, which can be further exacerbated by anti-inflammatory therapies.

## Figures and Tables

**Figure 1 viruses-16-00308-f001:**
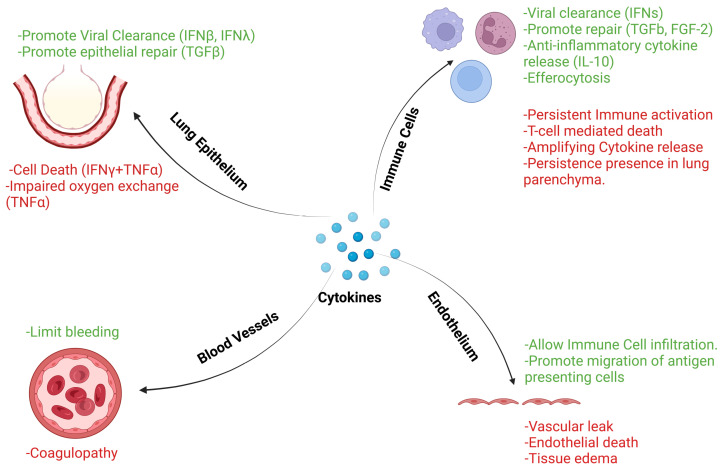
Beneficial and deleterious effects of cytokines on different cells indicate the complexity of targeting cytokines during clinical diseases. The green denotes potentially beneficial effects, while the red denotes potentially deleterious effects.

**Figure 2 viruses-16-00308-f002:**
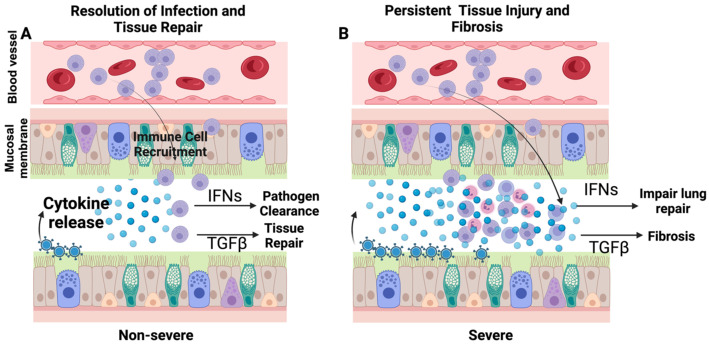
A viral pathogen is sensed by pattern recognition, which in turn releases the chemokines and cytokines to recruit inflammatory cells. In addition, some cytokines such as interferons directly prime the immune and structural cells to promote viral clearance by promoting the expression of interferon-stimulated genes. Upon pathogen clearance, the tissue damage is repaired by reparative cytokines such as TGFβ (**A**). In contrast, during severe disease, pathogen clearance is impaired, which leads to a massive release of cytokines and interferons leading to the accumulation of immune cells. At this stage, the elevated interferon levels may inhibit lung regeneration, and reparative cytokines like TGFβ lead to fibrosis (**B**).

**Figure 3 viruses-16-00308-f003:**
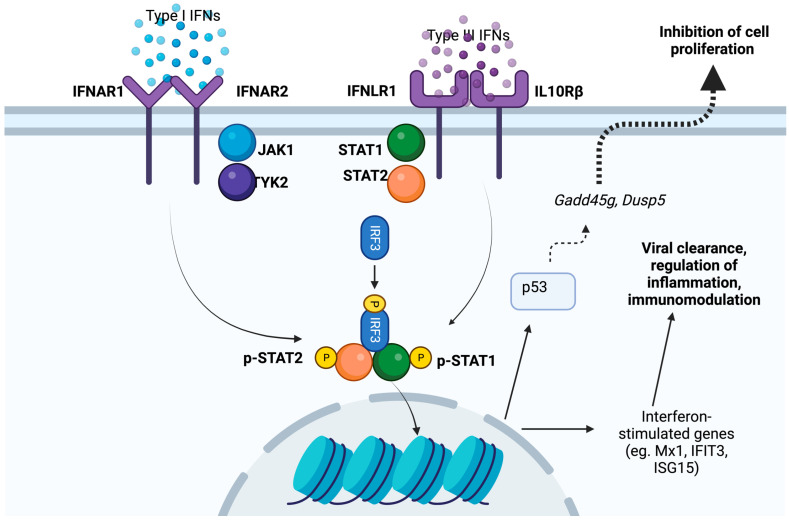
Mechanisms by which type I and type III interferons play beneficial and detrimental roles. These cytokines promote viral clearance by inducing hundreds of genes commonly called interferon-stimulated genes (ISGs). In contrast to the beneficial effects, interferons also induce the expression of p53 to inhibit cell proliferation, which is essential to repair damaged lung epithelium following viral infections such as influenza and COVID-19.

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
