# Peer review of "Beneficial and Detrimental Effects of Cytokines during Influenza and COVID-19"

_viruses, 2024, doi:10.3390/v16020308_

Round 1
Reviewer 1 Report
Comments and Suggestions for Authors
The authors have summarized cytokines’ diverse functions during influenza and COVID-19. This review has very good information about the beneficial and pathogenic effects of cytokines upon respiratory virus infections. Comments are made to increase the quality of the manuscript.
1. Please proofread the manuscript to correct errors. For example, injured tissue injury, IFNb, TGFb, we going to, ytokine, etc.
2. Figures: A. Figures are not referred in the text. B. Figure 1: It would be informative to write viruses (influenza or SARS-CoV-2) when the functions are written on the figure to indicate which virus was reported to have the cytokine’s effect on disease. C. Figure 3: IFN lambda receptor is IFNLR1, not IFNAL1.
3. Type I IFNs have diverse immune regulatory activity as well as antiviral function. This is not depicted in the Figure 3.
4. The role of IL-1b and inflammasome during influenza and COVID was mentioned, but not detailed.
5. In the title, does “Determinantal Effects” mean detrimental or something else?
Comments on the Quality of English Language
Okay
Author Response
Reviewer 1
Comments and Suggestions for Authors
The authors have summarized cytokines’ diverse functions during influenza and COVID-19. This review has very good information about the beneficial and pathogenic effects of cytokines upon respiratory virus infections. Comments are made to increase the quality of the manuscript.
Response: We really want to thank you for the insightful comments to improve the quality of the manuscript.
- Please proofread the manuscript to correct errors. For example, injured tissue injury, IFNb, TGFb, we going to, ytokine, etc.
Answer 1: Thanks a lot. We have proofread the manuscript.
- Figures: A. Figures are not referred in the text. B. Figure 1: It would be informative to write viruses (influenza or SARS-CoV-2) when the functions are written on the figure to indicate which virus was reported to have the cytokine’s effect on disease. C. Figure 3: IFN lambda receptor is IFNLR1, not IFNAL1.
Answer 2: Thanks for your comments. Firstly, we refereed the figures in the text. Secondly, didn’t write the name of specific viruses because these effects are shared by multiple viruses. Thirdly, in Figure 3, we have revised IFNAL1 into IFNLR1.
- Type I IFNs have diverse immune regulatory activity as well as antiviral function. This is not depicted in the Figure 3.
Answer 3: Thanks for your comments. We depicted it in the modified Figure 3.
- The role of IL-1b and inflammasome during influenza and COVID was mentioned, but not detailed.
Answer 4: Thanks for your comments. We have added it in the modified manuscript.
- In the title, does “Determinantal Effects” mean detrimental or something else?
Answer 5: Sorry for the mistakes. It’s “Detrimental”, we have corrected.
Reviewer 2 Report
Comments and Suggestions for Authors
The authors outline the beneficial and pathological effects of cytokines in influenza and COVID-19. It consists of the burden of respiratory viral infections, cytokines during respiratory infections, role of cytokines during respiratory infections, beneficial effects of cytokines, complexity in interpreting cytokine responses, cytokine inhibition, and objective assessment of deleterious effects of cytokine storm.
This explores the mechanisms by which SARS-CoV-2 induces the pathology through cytokine functions, which will be useful in understanding the pathogenesis and developing treatments for patients. However, there are some points that need to be reconsidered.
Subsection 2. The burden of respiratory viral infection: As the title of the paper indicates, influenza virus and SARS-CoV-2 are the main viruses addressed in this review. Both viruses show similarities in characteristics of respiratory tract infection, but differ in their structures and infectious cycles. At least, structural differences between the two viruses, such as segmented negative strand single RNA genomes and positive strand single RNA genomes, should be described in this subsection.
SARS-CoV-2 variants has different characteristics from the parental virus. Comments are needed on whether the omicron variant can be an inducer of cytokines as observed in alpha and delta variants.
4.1.1. Acute pathogenic roles of cytokines, line 145-147: The authors stated that “Of significance, a large proportion of severe patients in both influenza and COVID-19 are those with secondary bacterial infections”. Co-infection with influenza and SARS-CoV-2 has also been reported. This should also be discussed. An example of a related paper:
The host response to influenza A virus interferes with SARS-CoV-2 replication during coinfectionJ Virol. 2022 Aug 10;96(15):e0076522. doi: 10.1128/jvi.00765-22.
Page 2, line 89: IFNb should be IFN beta.
Page 6, line 254-256: Sun et al (ref. 58) stated that “an absence of IL-10 at the time of primary infection leads to enhanced local virus-specific antibody production”. This should be included.
Page 6, line 265: ytokine should be Cytokine.
Page 6, line 268-269: Denny eta l. (ref. 61) stated that “epithelial derived TGF beta acts to suppress early IFN beta responses leading to increased viral burden and pathology”. This is included to explain the diversity of TGF beta functions in influenza.
Page 7, lines 340-342: Tocilizumab is used to treatment patients with rheumatoid arthritis. Side effects of tocilizumab observed in this clinical use should be described.
Page 8, line 357-359: It is important to predict which cells produce these cytokines, i.e., TNF alpha, IL-16, and IL-1beta. The contribution of pro-inflammatory macrophages should also be included (ref. 80).
Figure 2: In the figure, the position of the blood vessels and respiratory mucosa should be shown.
Author Response
Review 2:
The authors outline the beneficial and pathological effects of cytokines in influenza and COVID-19. It consists of the burden of respiratory viral infections, cytokines during respiratory infections, role of cytokines during respiratory infections, beneficial effects of cytokines, complexity in interpreting cytokine responses, cytokine inhibition, and objective assessment of deleterious effects of cytokine storm.
This explores the mechanisms by which SARS-CoV-2 induces the pathology through cytokine functions, which will be useful in understanding the pathogenesis and developing treatments for patients. However, there are some points that need to be reconsidered.
Response: We really want to thank you for the insightful comments to improve the quality of the manuscript. Please see our response below to your suggestion.
Subsection 2. The burden of respiratory viral infection: As the title of the paper indicates, influenza virus and SARS-CoV-2 are the main viruses addressed in this review. Both viruses show similarities in characteristics of respiratory tract infection but differ in their structures and infectious cycles. At least, structural differences between the two viruses, such as segmented negative strand single RNA genomes and positive strand single RNA genomes, should be described in this subsection.
Answer 1: Thanks for your comments. We have added it.
SARS-CoV-2 variants has different characteristics from the parental virus. Comments are needed on whether the omicron variant can be an inducer of cytokines as observed in alpha and delta variants.
Answer 2: Thanks. We have added it.
4.1.1. Acute pathogenic roles of cytokines, line 145-147: The authors stated that “Of significance, a large proportion of severe patients in both influenza and COVID-19 are those with secondary bacterial infections”. Co-infection with influenza and SARS-CoV-2 has also been reported. This should also be discussed. An example of a related paper:
Oishi K, et al. The host response to influenza A virus interferes with SARS-CoV-2 replication during coinfection. J Virol. 2022 Aug 10;96(15):e0076522. doi: 10.1128/jvi.00765-22IF: 5.4 Q2 .
Answer 3: Thanks for your comments. We have added it.
Page 2, line 89: IFNb should be IFN beta.
Answer 4: Thanks. We corrected it.
Page 6, line 254-256: Sun et al (ref. 58) stated that “an absence of IL-10 at the time of primary infection leads to enhanced local virus-specific antibody production”. This should be included.
Answer 5: Thanks. We corrected it.
Page 6, line 265: ytokine should be Cytokine.
Answer 6: Thanks. We corrected it.
Page 6, line 268-269: Denny eta l. (ref. 61) stated that “epithelial derived TGF beta acts to suppress early IFN beta responses leading to increased viral burden and pathology”. This is included to explain the diversity of TGF beta functions in influenza.
Answer 7: Thanks. We corrected it.
Page 7, lines 340-342: Tocilizumab is used to treatment patients with rheumatoid arthritis. Side effects of tocilizumab observed in this clinical use should be described.
Answer 8: Thanks. We described it.
Page 8, line 357-359: It is important to predict which cells produce these cytokines, i.e., TNF alpha, IL-16, and IL-1beta. The contribution of pro-inflammatory macrophages should also be included (ref. 80).
Answer 9: Thanks. We included it.
Figure 2: In the figure, the position of the blood vessels and respiratory mucosa should be shown.
Answer 10: Thanks. We revised it.
Round 2
Reviewer 1 Report
Comments and Suggestions for Authors
Addressed the prior concerns.
Comments on the Quality of English Language
N/A